# Horizontal Gene Transfer and Drug Resistance Involving *Mycobacterium tuberculosis*

**DOI:** 10.3390/antibiotics12091367

**Published:** 2023-08-25

**Authors:** Xuhua Xia

**Affiliations:** 1Department of Biology, University of Ottawa, Ottawa, ON K1N 9A7, Canada; xxia@uottawa.ca; Tel.: +1-613-562-5718; 2Ottawa Institute of Systems Biology, University of Ottawa, Ottawa, ON K1H 8M5, Canada

**Keywords:** antibiotic resistance, phylogenetic incongruence, horizontal gene transfer, refampin, isoniazid

## Abstract

*Mycobacterium tuberculosis* (Mtb) acquires drug resistance at a rate comparable to that of bacterial pathogens that replicate much faster and have a higher mutation rate. One explanation for this rapid acquisition of drug resistance in Mtb is that drug resistance may evolve in other fast-replicating mycobacteria and then be transferred to Mtb through horizontal gene transfer (HGT). This paper aims to address three questions. First, does HGT occur between Mtb and other mycobacterial species? Second, what genes after HGT tend to survive in the recipient genome? Third, does HGT contribute to antibiotic resistance in Mtb? I present a conceptual framework for detecting HGT and analyze 39 ribosomal protein genes, 23S and 16S ribosomal RNA genes, as well as several genes targeted by antibiotics against Mtb, from 43 genomes representing all major groups within *Mycobacterium*. I also included *mgtC* and the insertion sequence IS6110 that were previously reported to be involved in HGT. The insertion sequence IS6110 shows clearly that the Mtb complex participates in HGT. However, the horizontal transferability of genes depends on gene function, as was previously hypothesized. HGT is not observed in functionally important genes such as ribosomal protein genes, rRNA genes, and other genes chosen as drug targets. This pattern can be explained by differential selection against functionally important and unimportant genes after HGT. Functionally unimportant genes such as IS6110 are not strongly selected against, so HGT events involving such genes are visible. For functionally important genes, a horizontally transferred diverged homologue from a different species may not work as well as the native counterpart, so the HGT event involving such genes is strongly selected against and eliminated, rendering them invisible to us. In short, while HGT involving the Mtb complex occurs, antibiotic resistance in the Mtb complex arose from mutations in those drug-targeted genes within the Mtb complex and was not gained through HGT.

## 1. Introduction

*Mycobacterium tuberculosis* (Mtb) is enigmatic for having two seemingly incompatible traits. Mtb grows slowly, with a generation time of about 24 h [1,2,3]. It also features a low mutation rate of about 2.5 × 10^−10^/bp/day [4]. Such an organism is not expected to acquire drug resistance rapidly in response to antibiotic challenges, yet Mtb has readily evolved extensive drug resistance [5,6,7] or even total drug resistance [5,8,9], leading to untreatable tuberculosis [8]. In a study monitoring 141 tuberculosis (TB) patients suffering from infection with Mtb strains that are susceptible to second-line drugs (SLDs), acquired resistance in Mtb was observed in 19 patients (14%) [10]. This rate of acquiring drug resistance is comparable to that in *Streptococcus pneumoniae* [11], which has a much shorter generation time [12] and a higher mutation rate [13,14]. 

One explanation for the rapid acquisition of drug resistance in Mtb is that drug resistance may evolve in other fast-replicating mycobacteria and then be transferred to Mtb through horizontal gene transfer (HGT). However, there has been little evidence of recent HGT. The lack or rarity of HGT has been attributed to three factors: (1) the thick cell wall that prevents exogenous DNA from entering the cell; (2) the lack of plasmids to transmit DNA across species; and (3) the limited repertoire of restriction enzymes and secretion systems to facilitate HGT. For these reasons, HGT is often discounted as a main mechanism for the acquisition of drug resistance [7,15,16,17,18]. Barring the possibility that drug resistance may arise in rapidly replicating mycobacteria and then transferred to slowly replicating mycobacteria, drug resistance in Mtb is often attributed to mutations within the Mtb complex, enhanced by genetic variation generated through insertion sequences [19,20,21] and fusion proteins [16], although genes responsible for antibiotic resistance are not associated with insertion sequences in the genomes of mycobacteria species [22].

The possibility that HGT contributes to drug resistance in Mtb, however, has neither been strongly rejected nor conclusively supported. Any HGT event involves a donor genome and a receptor genome. It is possible that previous studies may have missed either the donor genome or the receptor genome, or both, and consequently did not have a sufficient chance to detect rare evolutionary events such as HGT.

Two complementary phylogenetic methods have often been used for detecting HGT (and subsequent recombination) in viral and bacterial genomes. The first is based on phylogenetic incongruence (Figure 1A), illustrated numerically in Xia [23] (pp. 36–38). It addresses the question of which gene might be involved in horizontal gene transfer. The method needs a species tree (Figure 1A) and a gene tree (Figure 1B). The species tree is typically approximated by using a set of informational genes [24], i.e., those involved in genome replication, transcription, and translation that are less frequently involved in HGT and contain stronger phylogenetic signals than other genes. Single-copy genes, including most ribosomal protein genes are rarely observed in HGT [25,26], and the concatenated ribosome genes should provide an excellent approximation to the species tree. For cross-validation, one may also construct a tree from concatenated large ribosomal protein genes and another tree from concatenated small ribosomal protein genes. The gene tree is based on the gene suspected to be involved in HGT. If a gene is frequently involved in HGT, then its gene tree will tend to be incongruent with the species tree. In the gene tree in Figure 1B, species S4 gained gene 4 and 5 from species S2, so the phylogenetic position of S4 in the gene tree (Figure 1B) is incongruent with that in the species tree (Figure 1A). 

I should emphasize that, although a gene frequently involved in HGT tends to have a gene tree incongruent with the species tree, the phylogenetic incongruence itself is not evidence of HGT because it can also result from factors other than HGT, e.g., gene duplication followed by lineage sorting [27,28,29,30]. Suppose Mtb variants *microti*, *bovis*, and *africanum* share an identical version of gene X, but Mtb variant *bovis* also has a highly diverged homologue of gene X. The two versions of gene X in Mtb variant *bovis* could be either paralogues (through gene duplication) or xenologues (through HGT). Further evidence is needed to discriminate between the two.

The second phylogenetic method for detecting HGT and recombination is the SimPlot/DistPlot (similarity or distance plot) method, numerically illustrated in Xia ([23], pp. 49–53). The DistPlot method (Figure 1C–E) is more informative than the SimPlot method because the latter does not correct for multiple hits, whereas the former can use commonly used substitution models to correct for multiple hits [29]. DistPlot aims to accomplish two tasks. The first is to identify species that may be the donor or recipient in an HGT event (optionally with subsequent recombination). The second is to characterize which genes are involved in HGT. Figure 1C–E illustrates the scenario of six species (S0 to S5) with eight genes (1 to 8). Without HGT (and other factors distorting phylogenetic signals), the evolutionary distance between any species and other species is expected to be relatively constant. For example, given T_S_ in Figure 1A, the distance between species S5 and S4 ($d_{4,5}$) should be smaller than $d_{3,4}$, and this relative magnitude should be true for any genes, as illustrated in Figure 1C.

When HGT occurs with S4 gaining genes 4 and 5 from S2, then $d_{i,4}$ for genes 4 and 5 would differ from $d_{i,4}$ from other genes for all species descending from the common ancestor of the recipient S4 and donor S2 (Figure 1D). This change of $d_{i,4}$ suggests S4 as the recipient species. The HGT event has little effect on $d_{0,4}$ (Figure 1D) because S0 diverged before the common ancestor of S2 and S4. In contrast to the signature of the recipient (S4) in Figure 1D, the donor species, such as S2, does not change its distance to other species ($d_{i,2}$) except for $d_{2,4}$ (Figure 1E). This is because the donor’s phylogenetic position does not change except for its relationship to the recipient (S4). These patterns of pairwise $d_{i,j}$ values are important for inferring the donor and recipient species involved in HGT, assuming that the donor and recipient species, or at least their close relatives, have been included in the study. In most cases, the intermediate donor is a plasmid, while the bacterial donor could have already gone extinct and is therefore unidentifiable.

The DistPlot method is often applied with a sliding window along aligned gene sequences or aligned viral genomic sequences, with the objective of testing whether the gene or the genome might have participated in recombination and consequent incorporation of a diverged gene segment from sources other than ancestral inheritance. This approach is frequently used in identifying recombination events in viral genomes ([23] pp. 49–53).

The conceptual framework of inference has not been fully applied to the study of HGT in mycobacteria. In this paper, my main focus is on whether any species in the Mtb complex has served as a recipient species for a subset of genes that have been targeted by antibiotics. I approximated a species tree by a phylogeny based on ribosomal protein genes, and used phylogenetic incongruence and the DistPlot to detect the presence of HGT in the evolution of *Mycobacterium* lineages.

## 2. Results

### 2.1. Species Tree: Approximation from Ribosomal Protein Genes

There is excellent concordance between the tree derived from the 21 RPL proteins (Figure 2A) and that from the 18 RPS proteins (Figure 2B). Only three species (colored in red in Figure 2) differ slightly in their phylogenetic positions. Four distinct phylogenetic clades were labeled groups 1 to 4 (G1 to G4), shaded in blue, green, gray, and pink, respectively (Figure 2). The most striking feature of the two trees is the genetic homogeneity in the Mtb complex (*M. canettii* and *M. tuberculosis* variants *africanum*, *microti*, *bovis*, and H37Rv) and their clear separation from the rest of the *Mycobacterium* species.

G1 (Figure 2) consists of culturable, relatively fast-growing, but phenotypically and genetically diverse *Mycobacterium* species. Because of this diversity, two new genera, *Mycolicibacterium* and *Mycobacteroides*, have recently been proposed to accommodate them [31]. A subsequent morphological study [32,33] and a large-scale genomic comparison [34] supported the division. However, controversies with this taxonomic change remain [35]. While genetically diverse, as manifested by the long branches, these species are consistently grouped in the same clade in both the RPL tree (Figure 2A) and the RPS tree (Figure 2B). I will continue to refer to all of them as *Mycobacterium* species. 

G1 includes three nonpathogenic species. *M. dioxanotrophicus* was isolated from river sediment and capable of using 1,4-dioxane as a sole source of carbon and energy [36]. *M. aurum* was first isolated from a wastewater treatment plant in Prague and has the potential to degrade morpholine, which is an industrial waste [37,38]. *M. smegmatis* is best known for being used as a model organism for the Mtb complex [39,40]. G1 also includes *M. grossiae*, which was occasionally found in the human respiratory tract [41], and *M. abscessus*, which can cause severe diseases in immunocompromised patients and prompted the development and refinement of phage therapies because of its multi-drug resistance [42,43]. 

The five species in G1 also share a number of other unique features. For example, they all share a 3-codon insertion close to the 3′ end of the *rpoA* gene (encoding the α subunit of the RNA polymerase). However, I am interested only in features that lead to phylogenetic abnormalities implicating HGT and will not elaborate on features that are consistent with the putative species tree.

The rest of the species are slow-growing, with *M. leprae* having the longest generation time [2,44]. One would have expected the fast-growing G1 group to have much longer branches than the rest of the *Mycobacterium* species, and that the branches leading to the slow-growing *M. leprae* would be particularly short. However, this expectation is not supported (Figure 2).

G2 species are also phenotypically and genetically diverse, including a methanotrophic species found in sulfur caves (*M. sp_*sulfer_cave in Figure 2) with the tentative name of *Candidatus Mycobacterium methanotrophicum* [45], *M. cookie* isolated from sphagnum vegetation and surface water of moors in New Zealand [46] and occasional human pathogens such as *M. paraterrae* [47]. G3 is the *Mycobacterium avium complex* responsible for most of the nontuberculosis mycobacterial infections [48,49]. They can grow up to a temperature of 38.5 °C, which enables them to infect humans. Moreover, while a high fever can arrest the growth of these bacteria, it is insufficient to kill them because bacteria in the *Mycobacterium avium* complex can often survive up to 49 °C. G4 species includes the Mtb complex and the *M. leprae* complex, as well as many slow-growing *Mycobacterium* species causing non-tuberculosis infections. Some species, such as *M. shinjukuense*, can cause severe pulmonary disease [50]. 

Given that many nontuberculosis mycobacterial species are also found in human respiratory tracts and lungs, and are often isolated from tuberculosis patients, chances appear to exist for them to exchange genetic materials with the Mtb complex. However, the Mtb complex exhibits little genetic variation within the complex, but is well separated from the rest of the *Mycobacterium* species, with a branch length of 0.0907 in the RPL tree and 0.0792 in the RPS tree (Figure 2). This suggests that ribosomal protein genes are unlikely to be involved in HGT, confirming ribosomal protein genes as excellent phylogenetic signals for reconstructing the species tree.

Ribosomal proteins in mycobacteria have not been used as drug targets in developing antibacterials against the Mtb complex. Oxazolidinones bind to the large subunit ribosome and inhibit translation initiation by preventing the proper positioning of the initiator tRNA at the A-site [51,52]. However, the effect is more likely mediated by rRNA than ribosomal proteins for two reasons [51]. First, although binding studies showed interaction with the 50S ribosomal subunit, crosslinking studies revealed interaction partners to be the 23S and 16S rRNAs. Second, oxazolidinone-resistant isolates were found to have specific mutations (G2447T and G2576T) on the 23S rRNA [51,52]. Thus, the drug target is rRNA instead of ribosomal protein. However, ribosomal proteins have been targeted in other pathogenic bacteria, and nonsynonymous mutations at ribosomal protein genes can confer drug resistance in other bacterial species. For example, amino acid replacements at site G70 of the large ribosomal L4 protein reduce the susceptibility of *Neisseria gonorrhoeae* to azithromycin [53]. Similarly, nonsynonymous mutations in ribosomal proteins L22 and L4 can alter the three-dimensional structure of the large ribosome subunit, leading to changes in *E. coli* susceptibility to erythromycin [54]. This later study highlights the potential cost of drug resistance to bacterial species—mutations leading to an alteration of the three-dimensional structure of the ribosome are almost always deleterious. Thus, a foreign ribosomal protein or rRNA gained through HGT is most likely deleterious, leading to reduced fitness and eventual elimination of the HGT recipient.

### 2.2. Genomic Integrity in the Mtb Complex as Revealed by Ribosomal Proteins

I computed the evolutionary distance based on the TN93 model [29,55] over a sliding window of 500 nt between Mtb H37Rv and the other 42 *Mycobacterium* species included in this study. The 21 RPL genes and the 18 RPS genes from Mtb variants *microti*, *bovis*, and *africanum* are essentially identical to those in Mtb H37Rv. *Mycobacterium canettii* exhibits very small distances from Mtb H37Rv variant, as indicated by the red arrow in Figure 3. There is no indication of the Mtb complex gaining any RPL or RPS genes from the other *Mycobacterium* species included in this study, although it is possible that species not in the Mtb complex may participate in HGT.

The peak distance in RPL genes (Figure 3A) is due to indels in the 3′ half of the *rplQ* gene and the start of the *rplR* gene that render alignment difficult. Similarly, the peak distance in the RPS genes (Figure 3B) is caused by many indels in the *rpsP* gene, creating alignment ambiguities. However, all fluctuations in evolutionary distances among ribosomal proteins do not obscure the pattern that the genomes in the Mtb complex are highly homogenous, with little indication of HGT between the Mtb complex and other *Mycobacterium* species. 

### 2.3. Genomic Integrity in the Mtb Complex as Revealed by Ribosomal RNA Genes

Small subunit (ssu) ribosomes and the initiation tRNAs decode translation initiation signals in bacteria such as the Shine-Dalgarno sequence [56,57,58,59,60] and the start codon. Slow-growing *Mycobacterium tuberculosis* and *M. leprae* have only one 16S rRNA gene in their genomes, whereas fast-growing *M. smegmatis* features six. The still-faster-growing *Escherichia coli* (NC_000913) has not only seven 16S rRNA genes, but these genes also feature the strongest transcription signals at the −10 and −35 promoter sites, e.g., the *rrnB* gene in *E. coli* [61]. Ribosomal protein concentration is nearly perfectly correlated with growth rate in bacteria [62,63], highlighting the importance of rRNA availability in bacterial cells. 

Because translation machineries are essential for any form of life, they are often targeted by antibacterial drugs. Aminoglycoside, which binds to the A-site of the ssu ribosome to inhibit translation, is one such drug against pathogens in the Mtb complex [64]. However, drug resistance has been reported in many Mtb isolates [65]. Similarly, oxazolidinones bind to the large subunit ribosome and inhibit translation initiation by preventing the proper positioning of the initiator tRNA at the A-site [51,52]. However, specific mutations (G2447T and G2576T) on the 23S rRNA [51,52] lead to drug resistance. It is interesting to learn if rRNA genes in the Mtb complex exhibit signatures of HGT. Point mutations at rRNA genes leading to antibiotic resistance have also been observed in other bacterial pathogens, such as *Neisseria gonorrhoeae* [66,67].

Multiple copies of rRNA genes within each *Mycobacterium* species are nearly identical and are always tightly clustered together in a phylogenetic tree. Therefore, only one copy of the gene was used to represent each species in Figure 4A. The 23S and 16S rRNA genes were individually aligned and then concatenated with DAMBE. The phylogeny from this concatenated alignment (Figure 4) is largely concordant with the RPL tree (Figure 2A) and the RPS tree (Figure 2B). However, several differences are notable. First, the G2 clade is not monophyletic. Second, four species (colored red in Figure 4A) have changed their phylogenetic position relative to trees in Figure 2. Third, while the branch length leading to *M. abscessus* is comparable to those in Figure 2, the branch length separating the Mtb complex from the other species is only 0.0115, much smaller than those in Figure 2. This means that the genomes of the Mtb complex are not as genetically unique from the perspective of rRNA genes as they are from the ribosomal protein genes. In other words, rRNA genes in the Mtb complex may be involved in HGT.

The DistPlot output for the 23S rRNA gene (Figure 4B) shows TN93 distances along a sliding window of 500 nt between the Mtb H37Rv and other species. All members of the Mtb complex have nearly identical 23S rRNA genes. Only *Mycobacterium canettii* exhibits a slight variation (Figure 4B). If any member of the Mtb complex had gained an rRNA gene (or a segment of rRNA genes) from genetically diverged species, then its distance from the Mtb H37Rv would increase. However, this increase is small (Figure 4), suggesting that, if HGT occurred, the donor and the receptor should be very closely related. 

The only species with a segment of 23 rRNA identical to Mtb H37Rv is *M. shinjukuense* (red line in Figure 4B). However, the pattern suggests that it is *M. shinjukuense* that may have gained a segment from Mtb rather than the other way around, i.e., *M. shinjukuense* is a recipient rather than a donor in the putative HGT. *M. shinjukuense* is a recently identified pathogen that can cause severe pulmonary diseases [50]. Given the discovery of a plasmid that can shuttle information among slow-growing *Mycobacterium* species [68], it is possible for *M. shinjukuense* to gain genes from the Mtb complex. 

Figure 4B omitted all fast-growing species because the distances between them and the Mtb complex never came close. This is consistent with the finding of the only plasmid in the slow-growing *Mycobacterium* species [68]. This plasmid is limited to the slow-growing mycobacteria species and cannot cross the boundary to fast-growing relatives.

The pattern in 16S rRNA is similar (Figure 4C). Several *Mycobacterium* species (*M. basiliense*, *M. ulcerans*, *M. marinum*, and *M. shottsii*) have a long segment of 16S rRNA identical to that in the Mtb H37Rv. This is not because the rRNA segment is particularly conserved because it differs much from the homologous segment in *M. shinjukuense* (Figure 4C). The similarity of the 16S rRNA segment is more likely due to a plasmid donating the segment to the Mtb complex and the four *Mycobacterium* species. However, the sequence difference is small overall, as one can see from the scale on the vertical axis, so the data set has limited power in detecting HGT events.

### 2.4. RNA Polymerase β and β′ Subunit (rpoB and rpoC)

Transcription is essential for life, so the RNA polymerase β subunit (*rpoB*) gene has been targeted by antibiotics such as rifamycin including rifampin (rifampicin), rifapentine, and rifabutin. The mechanism of action is through the binding of rifampin to the rpoB protein, leading to the inhibition of transcription initiation and elongation. This is substantiated by five lines of evidence. First, the β subunit is involved in the resistance of several antibiotics, including rifampicin [69,70]. Second, rifampicin binds tightly to *E. coli* RNA polymerase [71]. Third, mutations in the β subunit often change rifampicin resistance [70,72]. Fourth, when the β subunit of RNA polymerase from a rifampicin-sensitive strain of *E. coli* is replaced by the β subunit from a rifampicin-resistant strain, the resulting RNA polymerase is rifampicin-resistant; in contrast, when the replacement is in the opposite direction, the resulting RNA polymerase is rifampicin-sensitive [69]. Fifth, when a particular mutation in the β subunit in a rifampin-resistant *Mycobacterium smegmatis* LR223 strain was introduced into the rifampin-sensitive LR222 strain of *M. smegmatis*, the latter became rifampin-resistant [73], with MIC increasing from 25 micrograms/mL to 200 micrograms/mL.

Mycobacterial RNA polymerases contain β and β′ subunits in tandem *rpoBC* configuration. Both were extracted, aligned, and analyzed. The β and β′ phylogenies (Figure 5A and Figure 5B, respectively) are concordant with each other and similar to the trees in Figure 2. The Mtb complex is well separated from the rest of *Mycobacterium* species, with the branch length being 0.0809 for β and 0.0511 for β′ (Figure 5). The Mtb complex is homogeneous in *rpoB*, but *M. canettii* differs slightly from the rest of the Mtb complex in *rpoC* (Figure 5B).

The *rpoB* sequences within the Mtb complex are nearly identical to each other (Figure 6A), with TN93 distances all nearly 0 between the Mtb H37Rv and the other members of the Mtb complex. There is no indication that any member of the Mtb complex has gained either a *rpoB* or a segment of it from any other *Mycobacterium* species (Figure 6A).

The pattern for the β′ subunit (Figure 6B) shows a difference between *M. canettii* and the rest of the Mtb complex that have nearly identical β′ subunit sequences with their $d_{i.Mtb_{H37Rv}}$ essentially all equal to 0 (the red line nearly overlapping the horizontal axis (Figure 6B). The sites 738–1284 in *M. canettii* contain 33 nucleotide differences (with two being nonsynonymous) from the other members of the Mtb complex. It is likely that *M. canettii* gained this segment from other species through HGT. Previous experimental assays have shown *M. canettii* to be an efficient DNA recipient in HGT [74]. However, searching this segment against GenBank returned only those within the Mtb complex, so the putative donor of this segment remains unknown.

### 2.5. Genomic Integrity in the Mtb Complex as Revealed by inhA and katG Genes

The success of Mtb depends heavily on its mycolic acids, which form a thick shell that prevents antibiotics from reaching the cell membrane in the Mtb complex [75]. The mycolic acids serve two key functions in Mtb survival [76]: (1) they contribute to bacterial resistance to antibiotics, and (2) they contribute to bacterial evasion of the host immune system [77]. Once reaching the lungs, Mtb is phagocytosed by pulmonary macrophages [2,3,78]. The infected macrophage dies, and the clumps of Mtb are phagocytosed by other macrophages. Macrophages usually trigger the apoptosis pathway after engulfing parasites, killing themselves as well as the parasites they have engulfed. Mtb can not only survive macrophage apoptosis but also use macrophages (and some other immune cells) to build a granuloma that serves as a protected niche for Mtb to survive and proliferate inside the cell. The defensive fortress of Mtb is constructed mainly with mycolic acids as the key components [76,79], so genes essential for the synthesis of mycolic acids represent the Archilles’ heel of Mtb [75]. 

Isoniazid inhibits the synthesis of mycolic acids by inhibiting NADH-dependent enoyl-[ACP] reductase encoded by *InhA* that is essential for mycolic acid synthesis [80]. The drug is specific against *Mycobacterium* species because it requires activation by a catalase-peroxidase encoded by the *katG* gene in *Mycobacterium* genomes. The normal function of KatG is to catabolize peroxides and protect mycobacteria from the harmful effects of peroxides [81]. However, KatG also oxidizes isoniazid to form isonicotinoyl radicals that inhibit InhA activity essential for mycolic acid biosynthesis [82]. Resistance to isoniazid in *Mycobacterium* species can arise from mutations in *InhA* [80,83], mutations in *katG* [84,85,86], or the deletion of *katG* [85].

*InhA* and *katG* differ from previous genes in that they are not informational genes and are therefore more likely to participate in HGT. The two trees (Figure 7) indeed differ much from the topologies in Figure 2 reconstructed from ribosomal protein genes. However, the genetic homogeneity within the Mtb complex in these two genes is also obvious from the trees. The branch lengths within the Mtb complex are essentially zero, but the branch length separating the Mtb complex from the rest of the species is large, being 0.0614 for the *inhA* gene (Figure 7A) and 0.1035 for the *katG* gene (Figure 7B). Thus, there is no indication that these two genes in the Mtb complex have participated in recent HGT, although HGT involving these two genes might occur beyond the Mtb complex.

Three species have multiple versions of *katG* (colored in Figure 7B). Are they paralogues from gene duplication or xenologues from HGT? The distribution of *katG* copies on the tree (Figure 7B) can be explained feasibly by two gene duplication events followed by gene loss. The first gene duplication gave rise to paralogues P_1_ and P_2_. *M. smegmatis*, *M. dioxanotrophicus* and *M. aurum* retained both paralogues in their genomes, but one paralogue, say P_2_, was lost in the ancestor of the blue and green clades (Figure 7B). Similarly, paralogue P_1_ was lost in the ancestor of the pink clade (Figure 7B). The second gene duplication generated paralogues P_21_ and P_22_ (Figure 7B). However, only *M. smegmatis* and *M. dioxanotrophicus* retained both P_21_ and P_22_, whereas three gene loss events occurred independently in the ancestor of the pink clade, in *M. abscessus*, and in *M. aurum*. This gene-loss event appears to be ongoing. For example, among the three *katG* genes in *M. smegmatis*, one gene (corresponding to *M. smegmatis*_2 in Figure 7B) does not produce a functional protein, and the other two catalyze the decomposition of peroxides during the exponential and stationary phases, respectively [81]. Thus, the first gene is already a pseudogene and on its way to gene loss.

Note that one could also concoct a scheme of multiple HGT events to explain the distribution of gene copies. However, these HGT events would need to occur at different times between donor and recipient species in such a way that phylogenetic relationships are largely maintained.

### 2.6. Is mgtC Involved in HGT?

The *mgtC* is a virulence factor important for maintaining Mg^2+^ homeostasis [87,88,89]. It has been suggested previously to be involved in HGT [90]. However, it has not been examined critically in a phylogenetic framework, so it is it not clear if multiple versions of the gene in the Mtb complex represent paralogues or xenologues.

The phylogeny of aligned *mgtC* sequences (Figure 8) is compatible with three gene duplication events. The first gene duplication generated paralogues P_1_ and P_2_. The colored species retained both P_1_ and P_2_, but the uncolored (black) species shaded in blue have lost P_2_. The second gene duplication event gave rise to P_21_ and P_22_ (Figure 8), but only four species (*M. dioxanotrophicus*, *M. grossiae*, *M. abscessus* and *M. stomatepiae*) have retained both copies. *M. stomatepiae*_2 has become a pseudogene with a frameshifting mutation and is therefore on its way to complete loss. The third duplication must have occurred very recently, after the divergence of *M. canettii* from the rest of the Mtb complex (Figure 8). Two observations substantiate this recent duplication. First, not only are the two copies of *mgtC* identical within Mtb_H37Rv, Mtb_microti, M_bovis, and M_africanum, but all eight of them are also identical to each other. Second, *M. canettii* has only one copy of P_1_, suggesting that the gene duplication occurred after the divergence of *M. canettii* from the rest of the Mtb complex.

### 2.7. Strong Evidence of HGT Involving Insertion Sequence IS6110

Insertion sequences occur frequently in mycobacteria, and their restricted distribution is often used for typing mycobacterial species or subspecies [91,92,93,94]. Insertion sequence IS6110 was previously thought to be specific and consequently was recognized from early on as a diagnostic signal for the Mtb complex in epidemiological studies [95,96,97]. Because IS6110 exists in multiple copies in the Mtb complex, e.g., 16 copies in the reference genome of *Mycobacterium tuberculosis* H37Rv (NC_000962), their locations and the length of intervening sequences can be used to distinguish among variants within the Mtb complex until a homologue of IS6110 was found in *M. smegmatis* [19]. HGT was suggested for the presence of an IS6100 homologue in *M. smegmatis* [19]. However, because the IS6110 homologue in *M. smegmatis* is only 67% identical to the IS6110 in the Mtb complex [19], both ancient gene duplication and HGT could be invoked as valid hypotheses.

Here, I present strong evidence for HGT involving IS6110. The genomes of the Mtb complex feature 16 IS6110 sequences that vary slightly in sequence length (1355–1375 nt). They are numbered from 1 to 16, of which numbers 1–8, 10, 12–14 and 16 are identical. I used this sequence to query the 67 *M. smegmatis* genomes for highly similar sequences. Two *M. smegmatis* sequences are sufficient to make a strong case for HGT because they are essentially identical to the IS6110 sequences from the Mtb complex (Figure 9).

There must be two very recent HGT events that generate the phylogenetic pattern in Figure 9. The Mtb complex and *M. smegmatis* are highly diverged from each other, as one can see from all previous phylogenies. The pattern in Figure 9 cannot be explained by ancient gene duplication because the divergence between NZ_SITV01000081 and IS6110-15 and, or between NZ_SITX01000082.1 and the other three Mtb H37Rv sequences, would correspond to the high divergence between *M. smegmatis* and Mtb H37Rv. However, the sequences within each of the two clades are essentially identical, with hardly any divergence. 

## 3. Discussion

To facilitate discussion, I should first define the concept of horizontal transferability that was previously proposed but not explicitly defined [98,99]. Let *e* represent an HGT event involving a gene transferred from a donor to a recipient, $p_{e}$ be the probability of such an HGT event in a given period, and $p_{f}$ be the probability of the transferred gene establishing its function in the recipient. Horizontal transferability (HT) would be equal to $HT=p_{e}p_{f}$. 

Both $p_{e}$ and $p_{f}$ depend on a variety of factors. In species within the family Pasteurellaceae and the genus *Neisseria* [100,101,102], gene sequences featuring an uptake signal sequence (or a DNA uptake sequence) have higher $p_{e}$ than other sequences. For mycobacterial species, no such uptake sequences have been reported, and $p_{e}$ may be assumed to be similar among genes. Thus, HT will depend mainly on $p_{f}$. Previous studies [98,99] suggest that functionally important genes, especially those forming protein complexes with many other partners such as ribosomal proteins and rRNA in ribosomes, would have a smaller $p_{f}$ than functionally unimportant genes such as insertion sequences. While this study does not measure $p_{e}$ or $p_{f}$, the framework is useful for discussion.

The phylogenetic pattern of the insertion sequence IS6110 suggests frequent HGT even between highly diverged mycobacterial species, such as between *M. smegmatis* and the Mtb complex. Whether these HGT events would be visible to us depends on $p_{f}$, i.e., how strong natural selection is against genes acquired through HGT. The Mtb complex appears to be highly adapted [103] and cannot tolerate mutations in functionally important genes. Many drug-resistant strains involve only one or a few mutations in specific genes, yet such a slightly modified gene would result in a significant reduction in fitness. For example, drug-sensitive strains of the Mtb complex have a generation time of about 20 h, but the corresponding drug-resistant strains have a generation time of about 30 h [104]. Thus, both functionally important and unimportant genes may be involved in HGT, but those strains acquiring a gene homologous to a functionally important gene are likely eliminated efficiently by selection, so we do not observe such HGT events. This is consistent with the functionally important genes studied in this paper, such as ribosomal protein genes, rRNA genes, and polymerase. However, for functionally unimportant genes such as insertion sequences, natural selection is expected to be weaker, leading to a relatively higher $p_{f}$, i.e., a higher chance for us to observe the HGT events.

The detection of frequent HGT events through insertion sequences between the Mtb complex and *M. smegmatis* is highly relevant to the battle against the drug-resistant Mtb variants. Genomic comparisons show that many virulence factors are shared between the Mtb complex and environmental mycobacteria [105], suggesting the potential of their exchange through HGT. The process could also generate new pathogens. For example, PPE25 and PPE26 from Mtb, which are not present in *M. smegmatis*, can significantly enhance the survival of *M. smegmatis* inside mouse macrophages when the two genes were expressed in *M. smegmatis* [106]. Similar results have been reported for other Mtb proteins, such as SA-5K [107], PE17 [108], Erp [109], Rv0431 [110], and isocitrate lyase [111] that enhance the survival of *M. smegmatis* in macrophages or other host cells. Multiple iterations of HGT, recombination, and natural selection could potentially generate new pathogens.

The results of this study do show the rarity of HGT involving functionally important genes, presumably because of a low $p_{f}$, and this could eliminate certain risks of HGT. For example, Mtb features both rifampicin-susceptible and rifampicin-resistant strains, and *M. smegmatis* also features rifampicin-susceptible and rifampicin-resistant strains, e.g., the LR222 and L223 strains, respectively [73]. Given that rifampicin-resistant mutations typically occur in the *rpoB* gene, there are at least two possibilities. First, rifampicin-resistant mutations occur independently at *rpoB* in different lineages of Mtb and *M. smegmatis*. Second, rifampicin-resistant mutations occur in the *rpoB* gene only once in either an Mtb lineage or an *M. smegmatis* lineage and are acquired by the other lineage through HGT. This study, however, does not favor the second hypothesis because *rpoB* is a functionally important gene with its protein interacting with multiple partners. Incorporating a diverged variant into the Mtg genome is likely deleterious, so such a recipient is likely eliminated efficiently by natural selection. 

I wish to emphasize the point that phylogenetics can shed light on many biopharmaceutical problems. Here is one additional example. Suppose we have a gene tree with a fast-evolving clade and a slow-evolving clade (Figure 10). If this gene or its product is a drug target, then the drug target is good if the pathogen is within the conserved clade (e.g., S1, S2, or S3), but not good if the pathogen is in the fast-evolving clade (S4, S5, or S6). A fast-evolving clade implies a relatively rapid accumulation of mutations, which facilitate the origin of drug resistance. If the pathogen is in the slow-evolving clade, then it has a reduced chance of evolving drug resistance. Such a simple analysis is often not done with the selection of drug targets.

Comparison of phylogeny and DistPlot patterns among genes can also help refine the selection of drug targets. For example, given the DistPlot for ribosomal protein genes and that for rRNA genes, one would choose ribosomal protein genes over rRNA genes as drug targets, everything else being equal. Similarly, given the DistPlot for *rpoB* and *rpoC* genes, one would choose *rpoB* over *rpoC* as a drug target.

## 4. Materials and Methods

There are 101 *Mycobacterium* species with sequenced genomes in GenBank. However, only 41 are annotated as “complete”. These do not include species that were traditionally in *Mycobacterium* but have recently been assigned new generic names, such as *Mycolicibacterium smegmatis*, *M. aurum*, and *Mycobacteroides abscessus*. In this 41 genomes, *Mycobacterium tuberculosis* was represented only by the strain H37Rv. After including *Mycolicibacterium smegmatis*, *M. aurum* and *Mycobacteroides abscessus*, as well as four additional members within the Mtb complex (i.e., *M. microti*, *M. bovis*, *M. africanum*, and *M. canettii*), there are a total of 48 genomes. However, five genomes exhibit phylogenetic incongruence between the tree built with individually aligned and concatenated large ribosomal protein genes and that built with aligned and concatenated small ribosomal protein genes. I cannot decide if this is due to sequencing errors or genomic fluidity. For simplicity, I just removed these five genomes, leaving 43 genomes included in this study.

To approximate the species tree, I extracted 21 large ribosomal protein (RPL) genes and 18 small ribosomal protein (RPS) genes that are shared among all 43 species using DAMBE [112]. These genes were individually aligned with MAFFT [113], with the slow but accurate LINSI option. The 21 RPL genes were then concatenated to create an RPL43.FAS file. The 18 RPS genes were similarly concatenated to create an RPS43.FAS file. These two supermatrix files were included as Appendix A. Aligned and concatenated 23S and 16S rRNA genes from the 43 genomes are also included as a Appendix A. Maximum likelihood trees were reconstructed with PhyML [114] with the GTR + Γ substitution model and simultaneous optimization of tree topology, branch lengths, and model parameters.

Drug development typically targets genes with essential functions for pathogen survival and reproduction. Such genes include 23S and 16S rRNA genes, *rpoB*, *rpoC*, *inhA*, and *katG*. Two genes were previously reported to be involved in HGT: the *mgtC* gene [90] and the insertion sequence IS6110 [19], which were included in this study. Gene trees were reconstructed from these individual genes to check if they might participate in HGT. All these genes are present in all 43 *Mycobacterium* species included in this study.

For the DistPlot method, simultaneous estimated TN93 distances [29] were computed with a sliding window of 500 nt. The simultaneous estimation is more robust and accurate than the commonly used independent estimation [29,115]. The use of the TN93 distance instead of the more general GTR distance is because the window size of 500 nt often does not include all six different nucleotide replacements (two transitions and four transversions) needed for parameter estimation in the GTR model.

For identifying HGT between the Mtb complex and *M. smegmatis* involving the insertion sequence IS6110, I downloaded the 67 genomes available for *M. smegmatis*, created a local BLAST database, and searched the IS6110 sequences from the Mtb complex against *M. smegmatis* genomes. Matches could arise from HGT or ancient gene duplication followed by subsequent lineage sorting. However, if the IS6110 sequences in *M. smegmatis* and the Mtb complex are identical, then HGT is favored.

## 5. Conclusions

As revealed by insertion sequences, horizontal gene transfer (HGT) occurs frequently between *Mycobacterium tuberculosis* (Mtb) and other mycobacterial species, even between Mtb and the highly diverged *M. smegmatis*. This raises the possibility that drug resistance may evolve in fast-replicating and large populations of nontuberculosis mycobacteria and get imported into Mtg through HGT. Fortunately, the horizontal transferability of genes depends on the functional importance of the gene. Functionally important genes are typically well adapted with their interacting partners, so a recipient of a diverged homologue through HGT is most likely deleterious and would be eliminated efficiently by natural selection. Phylogenetic patterns characterized by genomic data suggest that functionally important genes are rarely involved in HGT, but functionally unimportant sequences, such as insertion sequences, are readily observed in HGT. Because genes that have been chosen as drug targets are typically functionally important genes, they do not seem to participate in HGT, so there is no evidence suggesting that drug resistance in Mtb is gained through HGT. The analysis also shows that *mgtC*, which was previously suggested to be horizontally transferred, has phylogenetic patterns more consistent with gene duplication and subsequent lineage sorting than with HGT.

## Figures and Tables

**Figure 1 antibiotics-12-01367-f001:**
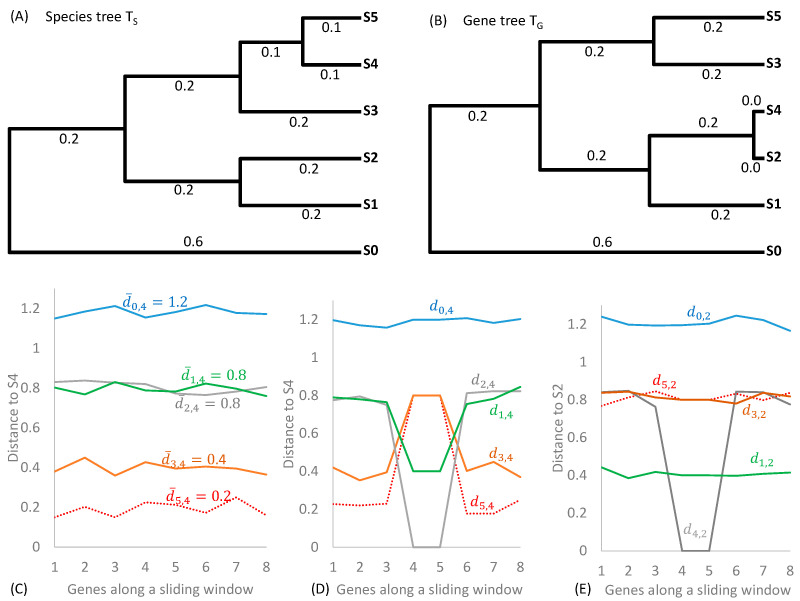
Phylogenetic methods for detecting horizontal gene transfer and recombination in viral and bacterial genomes. (**A**) A species tree T_S_. (**B**) A gene tree T_G_ resulting from species S4 horizontally acquiring the gene from S2. (**C**) Expected evolutionary distance (${\bar{d}}_{ij}$) between S4 and other species for eight genes (1 to 8), given T_S_. The two subscripted *i* and *j* in ${\bar{d}}_{ij}$ are species indicators. The observed rRNA distance may fluctuate stochastically above and below $\bar{d}$. (**D**) Distance $d_{i,4}$ between S4 and other species for the eight genes, given that species S4 gained genes 4 and 5 from species S2. For all species descended from the common ancestor of S2 and S4, $d_{i,4}$ would differ among genes because S4 gained genes 4 and 5 from S2 so T_G_ from these two genes would differ from T_S_. (**E**) Distance $d_{i,2}$ between S2 and other species for the eight genes. Only $d_{4,2}$ differs among genes, but other $d_{i,2}$ distances do not, because S2 does not change its phylogenetic positions.

**Figure 2 antibiotics-12-01367-f002:**
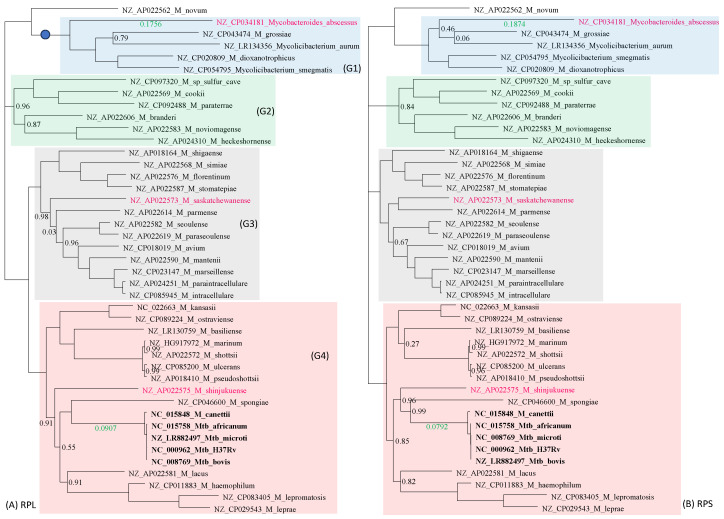
Approximation of the species tree with genes encoding ribosomal proteins. Species names are in the format of “GenBank accession”_”Species Name”. (**A**) Phylogeny from 21 aligned and concatenated RPL (large ribosome protein) genes. (**B**) Phylogeny from 18 aligned and concatenated RPS (small ribosomal protein) genes. The two trees are highly concordant, with only three species (colored red) differing slightly in their phylogenetic positions. The trees are unrooted. Bootstrap values equal to 1.00 are not shown, but those smaller than 1.00 are next to the node. The green-colored branch lengths, one leading to *Mycobacteroides abscessus*, and the other separating the Mtb complex from the rest, serve the function of a scale bar. Corresponding clades in (**A**,**B**) are shaded in the same color.

**Figure 3 antibiotics-12-01367-f003:**
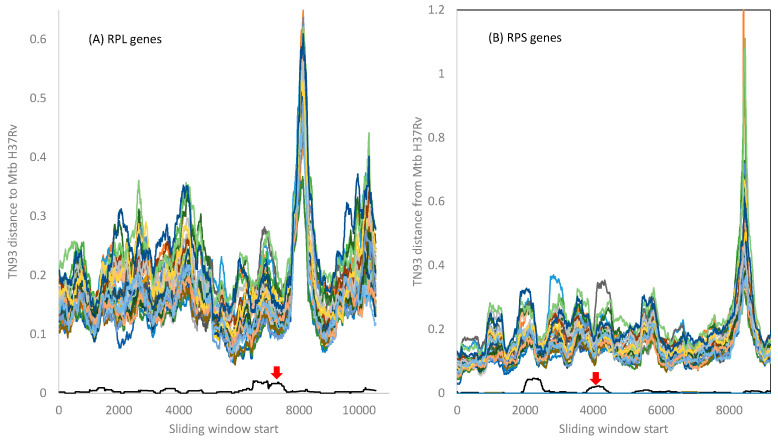
TN93 distances between the Mtb H37Rv strain and the other 42 species in Figure 2. The 21 RPL genes (and their aligned lengths) are rplA(720), rplB(843), rplC(702), rplD(735), rplE(600), rplF(540), rplI(462), rplJ(657), rplK(435), rplL(405), rplM(444), rplN(369), rplO(447), rplP(417), rplQ(702), rplR(414), rplS(342), rplT(396), rplU(324), rplV(780), and rplX(327). The 18 genes (and their aligned lengths) are rpsA(1467), rpsB(906), rpsC(873), rpsD(606), rpsE(777), rpsF(291), rpsG(471), rpsH(399), rpsI(543), rpsJ(306), rpsK(444), rpsL(375), rpsM(375), rpsO(270), rpsP(633), rpsQ(456), rpsS(282), rpsT(267). The red arrow points to the distance curve between Mtb H37Rv and *Mycobacterium canettii*. The distance between Mtb H37Rv and other *Mycobacterium* variants is effectively zero for all ribosomal proteins.

**Figure 4 antibiotics-12-01367-f004:**
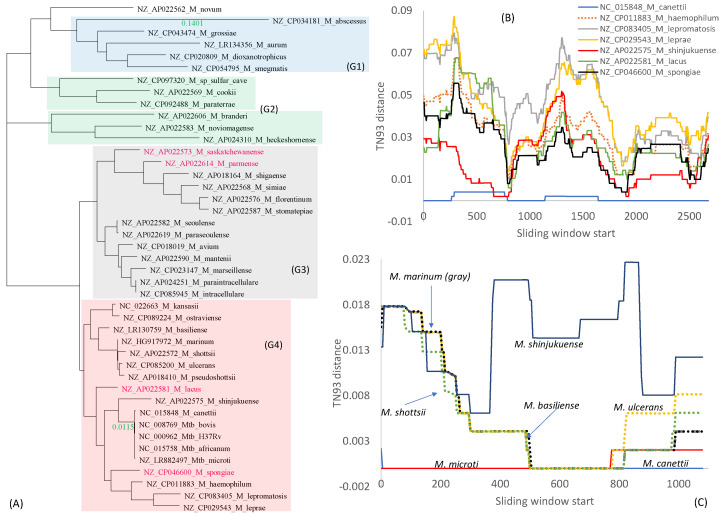
Genomic integrity as seen from rRNA genes. (**A**) Phylogeny from concatenated 23S + 16S rRNA genes. DistPlot of the 23S rRNA (**B**) and 16S rRNA (**C**) genes between Mtb H37Rv and its close relatives within the G4 clade. The G2 clade is not monophyletic as seen in Figure 2. Species that changed their phylogenetic position relative to tree in Figure 2 are colored red. The branch length to *M. abscessus* (colored green) is comparable to that in Figure 2, but the branch length (colored green) separating the Mtb complex from the rest is much shorter than that in Figure 2. TN93 distances were calculated over a sliding window of 500 nt.

**Figure 5 antibiotics-12-01367-f005:**
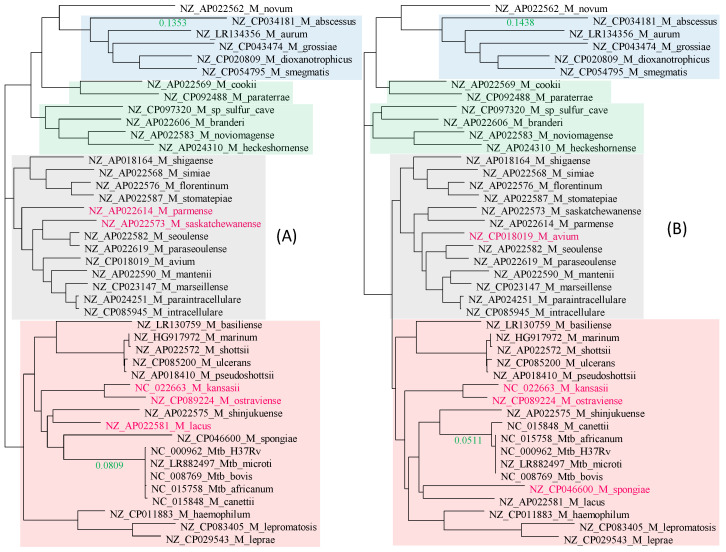
Phylogenetics of β (**A**) and β′ (**B**) subunits of RNA polymerase. Species with phylogenetic position different from those in Figure 2 are colored in red. The branch lengths (colored green) leading to *M. abscessus* and separating the Mtb complex from the rest were shown.

**Figure 6 antibiotics-12-01367-f006:**
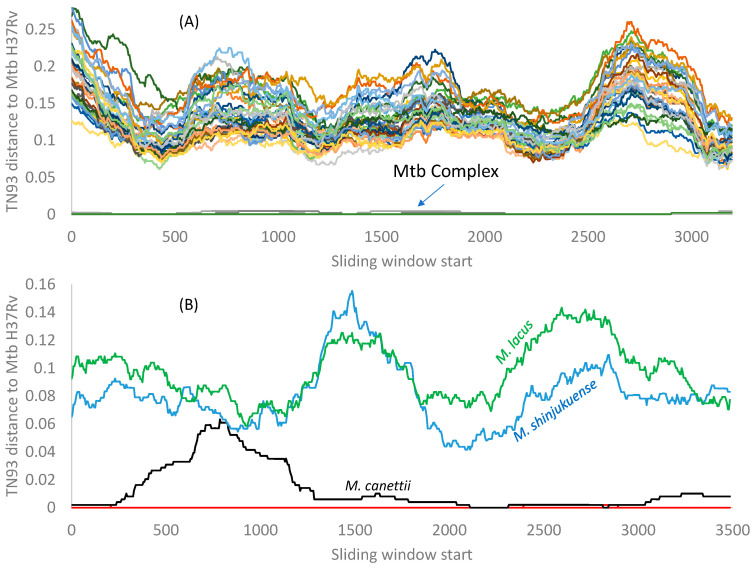
DistPlots for β (**A**) and β′ (**B**) subunits of the RNA polymerase in *Mycobacterium* species. TN93 distances were calculated between Mtb H37Rv and each of the other species over a sliding window of 500 nt. All members in the Mtb complex were colored red in (**B**) except for *M. canettii*, which was colored black.

**Figure 7 antibiotics-12-01367-f007:**
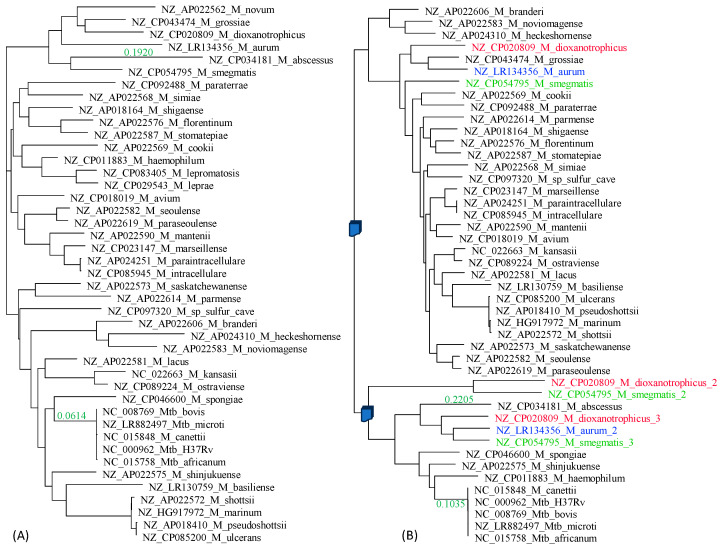
Phylogenies from *inhA* (**A**) and *katG* (**B**) genes. The branch lengths (colored green) leading to *M. abscessus* and separating the Mtb complex from the rest were shown next to the branch. Species with multiple versions of *katG* are highlighted with different colors, with an appended “_1”, “_2” and “_3” to distinguish different versions of the gene within the same genome. The numbering does not imply relationship.

**Figure 8 antibiotics-12-01367-f008:**
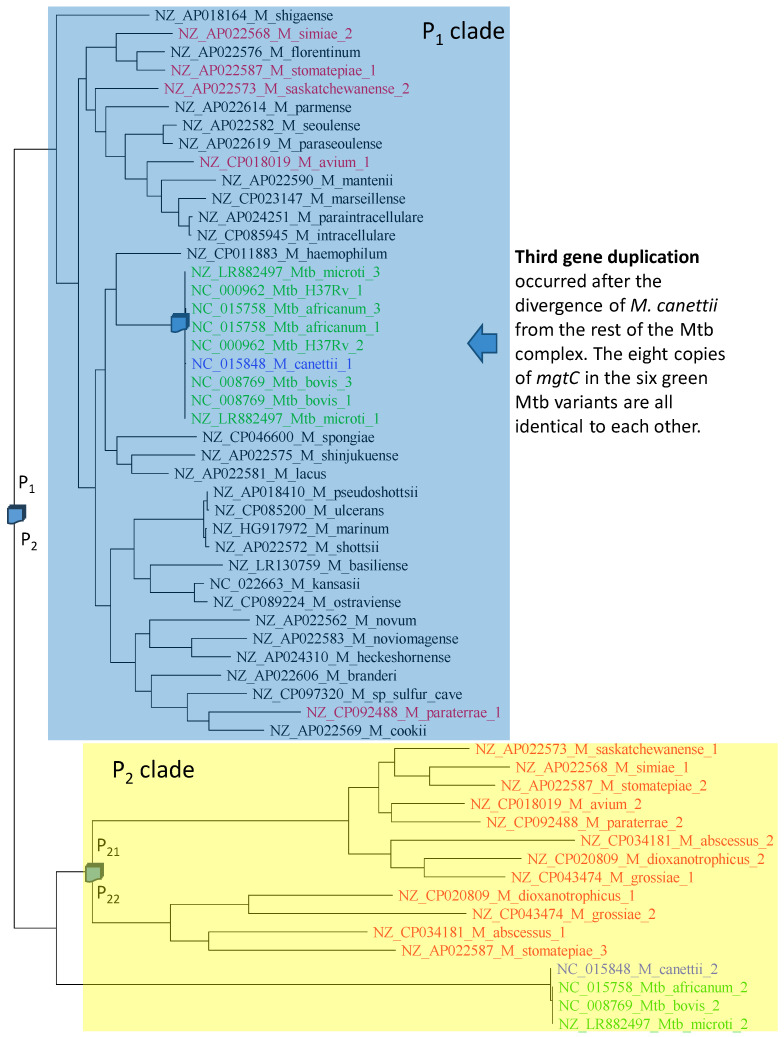
The phylogeny of *mgtC* is compatible with three gene duplication events. Species with multiple versions of *mgtC* are colored and appended with “_1”, “_2” and “_3” to distinguish different versions of the gene within the same genome. The numbering does not imply relationship. The Mtb complex is colored in green and blue. Other species with multiple versions of *mgtC* are colored red.

**Figure 9 antibiotics-12-01367-f009:**
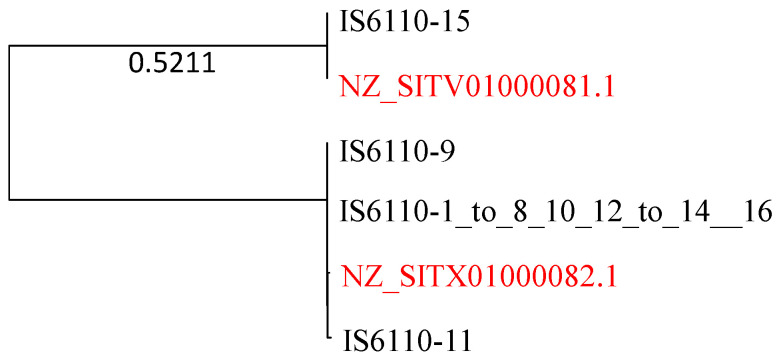
Phylogeny of the 16 IS6110 sequences from Mtb H37Rv (numbered from 1 to 16) and two homologues from *M. smegmatis* (colored red). The Mtb H37Rv sequences numbered 1–8, 10, 12–14 and 16 are identical and represented by one sequence. The tree is midpoint-rooted, with the number 0.5211 indicating the branch length.

**Figure 10 antibiotics-12-01367-f010:**
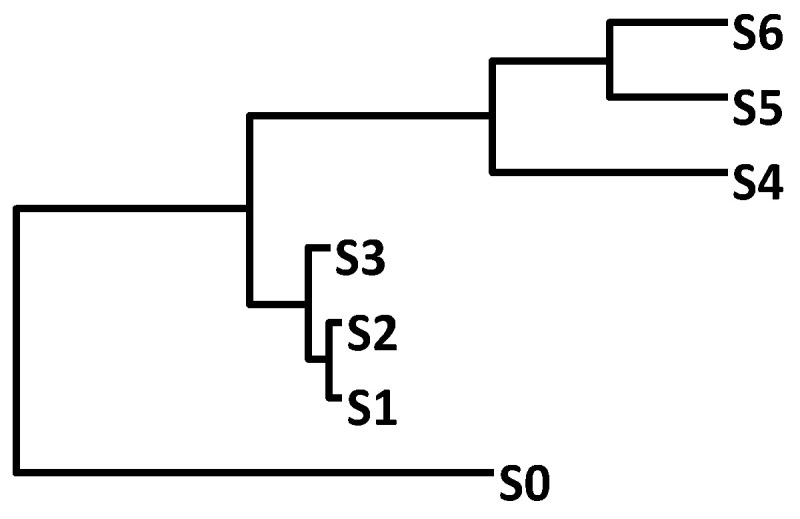
A tree with a fast-evolving clade (S1 to S3) and a slow-evolving clade (S4 to S6).

## Data Availability

Not applicable.

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
