# Peer review of "Horizontal Gene Transfer and Drug Resistance Involving Mycobacterium tuberculosis"

_antibiotics, 2023, doi:10.3390/antibiotics12091367_

Round 1

Reviewer 1 Report

Dear Author! You have presented an excellent work with an interesting scientific idea. The realization is done at a good level. To improve the work it is necessary to correct the references in the text (there is an error everywhere). In my opinion, this article is a good basis for further practical research.

Author Response

I have updated the references with incomplete information.

Reviewer 2 Report

The topic of this manuscript is very important. The author presented a framework for detecting HGT and analyzing ribosomal protein genes, ribosomal RNA genes, and other genes targeted by antibiotics against Mtb. The author is encouraged to address the following questions:

1. In the abstract, the text reads that 43 genomes were included in this study. In line 131, sequences from 43 Mycobacteria were downloaded for phylogenetic comparison. The key question is how the author chose these species and strains. What are the criteria for you to include this species and strains in this study? Did you consider the origin, the host, the phenotype, and drug resistance of these isolates?

2. This manuscript contains 10 figures and additional supplemental materials. This reviewer has difficulties reading the whole manuscript. While 43 genomes were used to generate ten figures, I have trouble identifying the take-home message from this study. Particularly, the author should add key findings in the abstract of this manuscript.

3. In this work, Mtb is one of the 43 Mycobacterium species included in this study. Therefore, the conclusion of this manuscript is not limited to Mtb. The author may consider removing "tuberculosis" from the manuscript title.

4. For each figure, the figure legend should be provided. In addition, the author often used the word "I", please consider switching to the passive tense. 

Author Response

(My response is in bold italic.)

The topic of this manuscript is very important. The author presented a framework for detecting HGT and analyzing ribosomal protein genes, ribosomal RNA genes, and other genes targeted by antibiotics against Mtb. The author is encouraged to address the following questions:

  1. In the abstract, the text reads that 43 genomes were included in this study. In line 131, sequences from 43 Mycobacteria were downloaded for phylogenetic comparison. The key question is how the author chose these species and strains. What are the criteria for you to include this species and strains in this study? Did you consider the origin, the host, the phenotype, and drug resistance of these isolates?

Good point. There are 101 Mycobacterium species with sequenced genomes in GenBank. However, only 41 are annotated as "complete". These do not include species that were traditionally in Mycobacterium but have recently been assigned to new generic names such as Mycolicibacterium smegmatis, M. aurum, and Mycobacteroides abscessus. Also, Mycobacterium tuberculosis was represented by the genome of the strain H37Rv only. After including Mycolicibacterium smegmatis, M. aurum and Mycobacteroides abscessus, as well as those within the Mycobacterium tuberculosis complex (i.e., M. microti, M. bovis, M. africanum, and M. canettii), there is a total of 48 genomes. However, five genomes exhibit phylogenetic incongruence between the tree built with aligned and concatenated large ribosomal protein genes and that built with aligned and concatenated small ribosomal protein genes. I cannot decide if this is due to sequencing error or genomic fluidity. For simplicity, I just removed them,  leaving 43 genomes included in this study.

I have included this information in the manuscript.

  1. This manuscript contains 10 figures and additional supplemental materials. This reviewer has difficulties reading the whole manuscript. While 43 genomes were used to generate ten figures, I have trouble identifying the take-home message from this study. Particularly, the author should add key findings in the abstract of this manuscript.

Excellent suggestion. I thought that I had made my point clear. It turned out that I did not. This paper aims to address three questions. First, does HGT occur between Mtb and other mycobacterial species? Second, what genes after HGT tend to survive in the recipient genome? Third, does HGT contribute to antibiotic resistance in Mtb? The main point is that horizontal gene transfer (HGT) occurs between the Mtb complex and other species. However, the fate of horizontal transferred genes differs. Functionally unimportant genes such as insertion sequences are not strongly selected against after HGT. They are visible on genomes participating in HGT. For functionally important genes such as ribosomal protein genes or those that have been chosen as drug targets, a diverged "foreign" homologue from another species is expected to be disruptive and would not work with other interacting partners as well as the home-adapted counterpart. For this reason, such genes, after HGT, would be strongly selected against and eliminated. They would disappear and become invisible. This interpretation is consistent with the phylogenetic patterns and substantiates a previously proposed hypothesis {Nakamura, 2004 #23655}{Creevey, 2011 #63524} that horizontal transferability of genes depends on their function. There is no evidence that the antibiotic resistance arising in the Mtb complex is due to HGT involving drug-targeted genes. The resistance is due to mutations within the Mtb complex.

I have revised the text to highlight the point.

  1. In this work, Mtb is one of the 43 Mycobacterium species included in this study. Therefore, the conclusion of this manuscript is not limited to Mtb. The author may consider removing "tuberculosis" from the manuscript title.

I have changed "in Mycobacterium tuberculosis" to "involving Mycobacterium tuberculosis"

  1. For each figure, the figure legend should be provided. In addition, the author often used the word "I", please consider switching to the passive tense. 

I did provide figure legends for the figures, but MDPI's reformatting of the manuscript seems to cause linked figure indices and reference indices unlinked, leading to many "Error! Reference source not found".

Reviewer 3 Report

The manuscript entitled ‘Horizontal gene transfer and drug-resistance in Mycobacterium tuberculosis’ by Xuhua Xia presents a conceptual framework for detecting HGT and analyzes various genes targeted by antibiotics against Mtb from 43 genomes. The author also explores the phylogenetic patterns of certain genes and the potential consequences of HGT involving both functionally important and unimportant genes.

The paper suffers from some major technical problems.

A more extensive and careful use of literature should be used along the manuscript since there are some sentences that does not include a reference. For instance, line 54-71, please re-read and check.

Line 57: in Xia (23, pp.??

Line 123

Error throughout the manuscript including figure legends: Error! Reference source not found

Author Response

(My response is in bold italic.)

The manuscript entitled ‘Horizontal gene transfer and drug-resistance in Mycobacterium tuberculosis’ by Xuhua Xia presents a conceptual framework for detecting HGT and analyzes various genes targeted by antibiotics against Mtb from 43 genomes. The author also explores the phylogenetic patterns of certain genes and the potential consequences of HGT involving both functionally important and unimportant genes.

The paper suffers from some major technical problems.

A more extensive and careful use of literature should be used along the manuscript since there are some sentences that does not include a reference. For instance, line 54-71, please re-read and check.

Added relevant references. Thanks for pointing out.

Line 57: in Xia (23, pp.??

This is a problem caused by MDPI's reformatting of the manuscript. It is pp. 36-38

Line 123

Error throughout the manuscript including figure legends: Error! Reference source not found

This is also a problem caused by MDPI's reformatting of the manuscript. I used Microsoft WORD's 'References' tools to index figures, and EndNote for index cited papers. These indices involve field codes in the manuscript. It seems that MDPI removed the code and then reformat the manuscript, leading to numerous "Error! Reference source not found". Reviewer #1 also has encountered these formatting problems.

I will inform MDPI of this problem. My submitted files in both WORD and PDF format do not have this problem.

Round 2

Reviewer 2 Report

I appreciate the efforts the author took to address all my concerns. I have no more comments. 

Reviewer 3 Report

The author justified and revised the work accordingly.